# Characterization of Papillomatous Lesions and Genetic Diversity of Bovine Papillomavirus from the Amazon Region

**DOI:** 10.3390/v17050719

**Published:** 2025-05-16

**Authors:** Fernanda dos Anjos Souza, Cíntia Daudt, André de Medeiros Costa Lins, Igor Ribeiro dos Santos, Lorena Yanet Cáceres Tomaya, Agnes de Souza Lima, Eduardo Mitke Brandão Reis, Rafael Augusto Satrapa, David Driemeier, Audrey Bagon, Cláudio Wageck Canal, Felipe Masiero Salvarani, Flavio Roberto Chaves da Silva

**Affiliations:** 1Laboratório de Virologia Geral e Parasitologia, Centro de Ciências Biológicas e da Natureza, Universidade Federal do Acre, Campus Universitário, BR 364, Km 04—Distrito Industrial, Rio Branco 69920-900, AC, Brazil; fernanda.anjos.asc@gmail.com (F.d.A.S.); agnes.lima@ufac.br (A.d.S.L.); edumitke@gmail.com (E.M.B.R.); rafael.satrapa@ufac.br (R.A.S.); audrey.bagon@ufac.br (A.B.); flavio.silva@ufac.br (F.R.C.d.S.); 2Laboratório de Medicina Veterinária Preventiva, Instituto de Medicina Veterinária, Universidade Federal do Pará, Castanhal 68740-970, PA, Brazil; andre.lins@castanhal.ufpa.br; 3Setor de Patologia, Faculdade de Veterinária, Universidade Federal do Rio Grande do Sul, Av. Bento Gonçalves, 9090, Prédio 42.505, Porto Alegre 91540-000, RS, Brazil; igor.ozn@gmail.com (I.R.d.S.); davetpat@ufrgs.br (D.D.); 4Centro de Ciências Exatas e Tecnológicas, Universidade Federal do Acre, Campus Universitário, BR 364, Km 04—Distrito Industrial, Rio Branco 69920-900, AC, Brazil; lorena.tomaya@ufac.br; 5Laboratório de Virologia, Faculdade de Veterinária, Universidade Federal do Rio Grande do Sul, Av. Bento Gonçalves, 9090, Prédio 42.505, Porto Alegre 91540-000, RS, Brazil; claudio.canal@ufrgs.br

**Keywords:** *Bos taurus*, *Papillomavirus*, bovine, papilloma, phylogeny, histopathology

## Abstract

Bovine papillomaviruses (BPVs) have been widely characterized from cutaneous warts in cattle worldwide. However, there are still limited studies addressing the geographic distribution of viral types and their potential associations with the histopathological characteristics of lesions, particularly in the vast and ecologically diverse Amazon region. This study aimed to histologically and phylogenetically characterize cutaneous papillomatous lesions in cattle from the Vale do Guaporé, located in the Brazilian Western Amazon. A total of 54 wart samples were collected from 44 cattle clinically diagnosed with cutaneous papillomatosis. Histopathological analysis classified 58.33% of cases as fibropapillomas and 39.58% as squamous papillomas. Molecular analysis, based on L1 gene amplification and sequencing, identified the presence of previously reported BPV types (BPV2, 4, 5, 12, 13, and 15), along with a novel BPV14 subtype and three putative new types (PNT). Statistical analysis revealed that BPV2 was significantly associated with fibropapillomas (*p* = 0.023), whereas BPV13 was linked to cauliflower-like morphological lesions (*p* = 0.008). These findings enhance the understanding of BPV diversity circulating in cattle from the Amazon region and provide valuable insights into the clinicopathological aspects of bovine cutaneous papillomatosis, which may aid in future epidemiological surveillance and disease control strategies.

## 1. Introduction

Papillomaviruses (PV) are small non-enveloped, oncogenic viruses measuring 55 to 60 nm in diameter. The viral capsid has icosahedral symmetry and presents 360 copies of the L1 protein. Their genome is composed of double-stranded circular DNA with around 8 kilo basis pair) [1]. Viruses from the Papillomaviridae family infect a diverse number of animals, including fish and a large number of amniotes, like birds, reptiles, and mammals, as well as humans. Papillomatosis is an infectious and contagious neoplastic disease, characterized by the presence of multiple benign tumors that can regress spontaneously or progress to malignant neoplasms, generally affecting the skin and mucous membranes of young and/or immunocompromised individuals [2,3]. Within this family, the *Human papillomaviruses* (HPV) are the most studied PVs [3] while *Bos taurus Papillomavirus* (bovine papillomavirus—BPV) are well known to infect the Cervidae, Giraffidae, and Bovidae families and are the second species researched beyond humans [2,3].

In cattle, bovine papillomavirus (BPV) infection can be asymptomatic or lead to the occurrence of papillomas and fibropapillomas on the skin and mucous membranes, neoplasms of the urinary bladder, and upper digestive tract [4,5]. Papillomaviruses are well known to cause stressful diseases in farm and companion animals [6]. Cutaneous papillomas in cattle are responsible for significant economic losses for beef and dairy livestock, according to the extent and anatomical location of the lesions [6]. The losses include reduced productivity and lower commercial value of animals and by-products such as leather, in addition to making milking difficult or impossible if located on the teats, predisposing to mastitis, myiasis, and bacterial infection, which can lead to the animal being discarded or to death [5,7,8]. Moreover, animals that develop persistent disseminated papillomas may have vision interference if located near the eyes [2,4,8]. Additionally, the upper GI cancer and urinary bladder on these hosts are commonly caused by chronic ingestion of bracken fern associated with concomitant infection of BPV [4,6].

Based on the degree of similarity of the nucleotide sequence of the L1 gene, biological properties and topology in the phylogenetic tree [9,10], BPVs are currently classified into 5 genus, 5 species, and 45 types [11]. The genus *Xipapillomavirus* comprises two species, *Xipapillomavirus 1* (BPV3, 4, 6, 9–11 and 15) and *Xipapillomavirus 2* (BPV12), in addition to *Xipapillomavirus* without classified species (BPV17, 20, 23, 24, 26, 28, and 29). In the genus *Deltapapillomavirus*, only one species is described in cattle, *Deltapapillomavirus 4* (BPV1, 2, 13, and 14), as well as the genus *Epsilonpapillomavirus*, with the species *Epsilonpapillomavirus 1* (BPV5, 8, and 25), and *Dyoxipapillomavirus*, with the species *Dyoxipapillomavirus 1* (BPV7). The genus *Dyokappapapillomavirus* (BPV16, 18 and 22) is not classified at the species level, while the types BPV19, 21, 27, 30–44, and tick-associated BPV are not classified at the genus level [11]. Although studies carried out in different Brazilian biomes have described several new BPV types, putative new types (PNT), and the occurrence of co-infections [12,13,14,15], the diversity of BPVs is still small when compared to the more than 400 types of papillomaviruses described in humans [11].

Generally, papillomaviruses are tissue-specific, with tropism for squamous epithelial cells and fibroblasts [4,5]. Thus, infection by *Deltapapillomavirus* leads to the transformation of subepithelial fibroblasts, followed by epithelial acanthosis and papillomatosis, comprising high risk BPV types for the development of malignant lesions, such as urinary bladder neoplasia. *Xipapillomaviruses* and *Dyoxipapillomaviruses* genus are considered epitheliotropic, responsible for the occurrence of papillomas, while the *Epsilonpapillomavirus* genus causes fibropapillomas and papillomas [2,4]. Co-infections and a possible association of epitheliotropic types with fibropapillomas are reported, as well as the occurrence of genotypes causing lesions in different anatomical regions [2,8].

The diagnosis of cutaneous papillomatosis can be made clinically, since the macroscopic changes are well characterized [2]. However, it is essential to perform molecular diagnosis as PCR once this technique is widely used due to its high sensitivity and specificity being applied even in situations where the amount of available DNA is reduced [16]. Furthermore, histological evaluation of the lesions allows classification into squamous papillomas or fibropapillomas, in addition to the identification of intraepithelial neoplasia, which may be associated with the oncogenic potential of the virus [4,16].

Therefore, the objective of this study was to analyze the possible association between viral types and the histopathological characteristics of cutaneous papillomatous lesions in cattle from the Brazilian Western Amazon region as well as the viral diversity circulating at the region studied.

## 2. Materials and Methods

### 2.1. Sample Collection

A total of 54 cutaneous lesions were collected from 44 beef (*n* = 22), dairy (*n* = 16) and mixed (*n* = 4) cattle, clinically diagnosed with papillomatosis based on macroscopic lesions. The samples were collected from 15 different farms (named I to XV) from the Vale do Guaporé region (constituted by the municipalities of São Miguel do Guaporé, São Francisco do Guaporé, Seringueiras and Costa Marques), State of Rondônia, Brazilian Amazon (Appendix A). Samples were collected by physical restraint of the animals followed by subcutaneous infiltrative local anesthesia by 2% lidocaine (Bravet, Rio de Janeiro, RJ, Brazil). Sterile scalpels and tweezers were used for each lesion. Afterwards, a fragment of each sample was stored at −20 °C until DNA extraction. For histopathological analysis, according to tissue availability, 48 fragments of these lesions were stored in 10% buffered formaldehyde. Data on age, sex, anatomical location of the lesions (head, neck or dewlap/thorax, back or abdomen/udder, groin or tail), and their macroscopic morphological characteristics for each animal were recorded.

To minimize animal suffering, all procedures were carried out in accordance with the European Convention for the Protection of Vertebrate Animals Used for Experimental and Other Scientific Purposes (2010/63/EU, revised 2010) and in accordance with Colégio Brasileiro de Experimentação Animal (COBEA). The project was approved by the Comissão de Ética no Uso de Animais da Universidade Federal do Acre (protocol number 23107.005499/2018-96). A schematic representation of the sample processing flow, including the inclusion and exclusion criteria at each analytical step, follows (Figure 1).

### 2.2. Macroscopic and Histopathological Analysis

The lesions collected were classified as cauliflower (irregular, with a cornified surface and a wide base of insertion), flat (dense and flat masses entirely connected to the tissue), pedunculated (connected by a narrow base of insertion), filamentous (masses with a thin base), and rice grain shape (small rice-shaped papillomas) [16].

Tissue samples were fixed in 10% buffered (50 mM phosphate, pH 7) formaldehyde for at least 72 h, subjected to routine histological processing, and stained with hematoxylin and eosin (HE) for microscopic analysis. The samples were evaluated individually and classified as squamous papilloma or fibropapilloma, according to the growth pattern and proportion of the epithelial layer and overlying dermal connective tissue [17,18]. In each case, the presence of increased intracytoplasmic keratohyalin granules, intranuclear eosinophilic inclusions, eccentric condensed nuclei with perinuclear halo (koilocytes), and cytoplasmic pallor/basophilia (viral cytopathic effect) in the proliferated epidermis were verified. If present, the inflammatory infiltrate was classified according to intensity (mild, moderate, and severe), cellular composition, and location.

### 2.3. DNA Extraction and PCR

Total DNA was isolated from papillomatous lesions using the PureLink^®^ Genomic DNA Mini Kit (Invitrogen, Carlsbad, CA, USA) and stored at −20 °C until use. Partial amplification of the L1 gene was performed with the degenerate primers FAP59 (5′-TAA CWG TIG GIC AYC CWT ATT-3′) (position in BPV1 X02346: 5712-5752) and FAP64 (5′-CCW ATA TCW VHC ATI TCI CCA TC-3′) (position BPV1 X02346: 6206-6185), resulting in an amplification product of 478 bp [19]. PCR was performed in a total volume of 25 μL, containing 2 μL of extracted DNA, 2.5 μL of 10X PCR buffer, 0.38 mM magnesium chloride, 1 unit of Taq DNA polymerase, 0.05 mM of each dNTP, and 0.2 μM of each primer. The reaction conditions were initial denaturation for 5 min at 95 °C, followed by 40 cycles of 94 °C for 1 min, 50 °C for 1 min, 72 °C for 1 min, and a final extension at 72 °C for 7 min. Aliquots of the reactions, stained with Loading Buffer Gel Red + Bromophenol + Xylenecyanol (Quatro G Biotecnologia, Porto Alegre, PA, Brazil), were subjected to electrophoresis in 2% agarose gels and visualized under ultraviolet light.

### 2.4. Sequencing and Phylogenetic Analysis

Amplicons were purified with the PureLink^®^ Quick PCR Purification Kit (Invitrogen, Carlsbad, CA, USA), and both strands were sequenced with the AB 3500 Genetic Analyzer (Applied Biosystems, Foster City, CA, USA) using a BigDye Terminator v. 3.1 Cycle Sequencing Kit (Applied Biosystems, Foster City, CA, USA). Sequencing data were collected using the Data Collection 3 program (Applied Biosystems, Foster City, CA, USA) and converted into FASTA files by Sequence Analysis Software v. 6 (Applied Biosystems, Foster City, CA, USA) according to standard parameters.

The sequences obtained in this study were edited using the Geneious software (version 2022.1.1) and then compared with the online database (GenBank 260.0) using the BLASTn tool [20]. All the BPV reference genomes, as well as the sequences with the greatest similarity to those obtained in this study were retrieved from NCBI (https://www.ncbi.nlm.nih.gov, accessed on 4 February 2025) for phylogenetic analysis.

Alignment of the sequences was carried out with Clustal W [21] and the phylogenetic tree was constructed by the maximum likelihood method (ML) inferred based on the best-fitting nucleotide substitution model, Tamura 3 parameter [22], with gamma distribution and invariant sites (T92+G+I), selected after using the “Find Best DNA/Protein Model” tool available in MEGA X (version 10.2.6) [23]. The reliability of the tree was tested with 1000 non-parametric bootstrap analyses. Bootstrap values > 50% were considered significant.

### 2.5. Statistical Analysis

Possible associations between BPV types and macro and microscopic characteristics of the samples subject to phylogenetic and histopathological analysis were analyzed using Fisher’s Exact Test (histological characteristic) [24] or Monte Carlo Simulation (anatomical site and macroscopic characteristics) [25] using R software (version 4.5.0) [26]. Significant associations were considered when *p* < 0.05.

## 3. Results

### 3.1. PCR, Sequencing, and Phylogenetic Analysis

Of the 54 samples collected, 51 (94.44%) were positive in the PCR protocol. These were sequenced and edited, resulting in 39 samples of high quality that were subjected to phylogenetic analysis. In the phylogenetic tree, the vast majority of BPVs identified belonged to the genus Deltapapillomavirus (32 sequences), followed by the genus Xipapillomavirus (3 sequences) and Epsilonpapillomavirus (1 sequence). Furthermore, three PNT associated with the Xipapillomavirus genus were identified (Figure 2).

The percentage at which associated taxa cluster together is shown next to the branches, bootstraps < 50% have been suppressed. Study sequences are shown in bold; black dots indicate putative new viral types (PNT). BPV2 was identified in 58.97% (23/39) of the samples, followed by BPV13 (17.96%; 7/39), BPV14 (5.13%; 2/39), BPV4 (2.56%; 1/39), BPV5 (2.56%; 1/39), BPV12 (2.56%; 1/39), and BPV15 (2.56%; 1/39).

Among the sequences that clustered into the *Deltapapillomavirus* genus, the samples classified as BPV2 (23) showed a high degree of identity between them (100%). The seven sequences that were grouped with BPV13 also showed 100% identity to the reference BPV13. The strains BRA38RO21 and BRA43RO21 also clustered with *Deltapapillomavirus* genus and were classified as putative new subtypes of BPV14 (95.42% and 95.70% identity, respectively).

Concomitant infection with BPV2 and 13 was detected in different lesions from the same animal from farm XIII (BRA25RO21a and BRA25RO21b). In addition, different viral types were identified in animals from farm X (BPV13 and a putative new PV type showing 78,08% of similarity with a previously described BPV38); farm XIII (comprising BPV2, 13, and 2 putative new type (PNT), one of them related to BPV42 (83.23% of similarity), and another related to a previously PNT of BPV described BPV/BR-UEL6 (91.98% of identity). Detailed information of each sample including properties, animal gender and age, anatomic sites of lesions, morphology and histopathology, sample identifications, and GenBank accession numbers can be accessed in the Appendix A.

In relation to the strains that clustered with the *Xipapillomavirus*, BPV15 (BRA06RO21), BPV4 (BRA41RO21), and BPV12 (BRA42RO21) were identified. In addition, three strains showed phylogenetic distances greater than (BRA33RO21, BRA17RO21) or around 10% (BRA26RO21) when compared to the sequences deposited in the online database. For this reason, the BRA17RO21, BRA26RO21, and BRA33RO21 strains were classified in this study as PNT. The BRA17RO21 and BRA26RO21 sequences (78.80% similarity between them) showed a common ancestor with BPV38 and BPV43. The BRA17RO21 strain showed 77.08% similarity with BPV38, while the BRA26RO21 strain showed 91.98% and 79.08% with the strain KP892554 and BPV15, respectively. The BRA33RO21 sequence showed a common ancestor with BPV42 (83.23% identity) and with the MZ292467 strain (84.19%). All results presented here are summarized in Figure 1 and Appendix A.

### 3.2. Histopathology and Macroscopic Characteristics of the Lesions

Macroscopically, the lesions clinically characterized as papillomatosis (54 samples) were classified as flat (59.26%) (Figure 3a), cauliflower (35.19%) (Figure 3b,c), and pedunculated (5.55%) (Figure 3d).

The histology of the 48 lesions analyzed revealed that 58.33% (28/48) were fibropapillomas, 39.58% (19/48) squamous papillomas, and 2.1% (1/48) were considered non-diagnostic, consisting only of hyperkeratosis. Fibropapillomas were characterized by discrete-to-moderate epidermal proliferation, forming elongations (rete pegs) towards the marked proliferation of fibroblasts in the overlying dermis. The proliferated fibroblast was arranged in random bundles and had discrete pleomorphism (Figure 4a). Squamous papillomas were formed by marked epidermal exophytic proliferation, forming filiform projections on the surface and supported by discrete dermal fibrous connective tissue (Figure 4b).

In both histological types, the epidermal proliferation was formed by hyperplasia of cells in the spinous stratum, covered by moderate to marked parakeratotic or orthokeratotic hyperkeratosis in the corneum stratum. Most cases showed an increase in intracytoplasmic keratohyalin granules (93.62%; 44/47) and the presence of an inflammatory infiltrate (46.81%; 22/47), characterized mainly by discrete to marked number of lymphocytes and plasma cells in the deep dermis or among the proliferated fibroblasts. There were occasional cases with koilocytes (27.66%; 13/47), intranuclear eosinophilic inclusions (14.89%; 7/47), and cytopathic viral effect (2.13%; 1/47).

Animal data, morphology, anatomical location and histology of the lesions, geographic location of the farms, as well as the types/PNT of BPVs found in the study and the respective accession numbers of the partial nucleotide sequences of the BPV L1 gene submitted to GenBank are shown in Appendix A.

### 3.3. Statistical Analysis

A total of 34 samples, which contained the results in phylogenetic and histopathological evaluation, could be used in the statistical analysis of the study. It was observed that BPV2 presented a significant frequency as fibropapilloma (*p* = 0.023). Furthermore, a potential trend toward association was observed between the presence of BPV13 and lesions with cauliflower morphology (*p* = 0.008) was also found. There was no significant association between the viral type and the anatomical sites of the lesions (head, neck, or dewlap/thorax, back or abdomen/udder, groin or tail), making it not possible to reject the null hypothesis (Appendix A).

## 4. Discussion

In this study, BPV DNA was amplified and sequenced in 51 of 54 cutaneous wart samples of beef, dairy, and mixed breed bovine from the Amazon region, State of Rondônia, Brazil. Additionally, different types of BPVs were identified in three known genera, using the degenerate primers FAP59/FAP64, as can be observed in other exploratory studies of this type [7,12,14]. Additionally, three PNT were identified in the present study, one of them previously described [15]. In the herds studied, BPVs 2, 4, 5, 12, 13 and 15 were identified, in addition to new subtypes of BPV14. In Brazil, many of these BPV types have already been described in cattle herds in the South [7,13], Southeast [27,28], Northeast [29] and North regions [13,30,31,32]. However, this is the first report to describe the presence of BPV4, 12, 14, and 15 in the Amazon region.

Although BPV4 is commonly found in the GI tract and is substantially associated with gastrointestinal carcinomas in synergism of bracken fern grazing [2], a study that analyzed 100 ruminal samples from cattle in the Western Amazon region described the fibropapilloma and squamous papillomas caused by BPV2 and 13 and squamous papilloma caused by BPV44 [32]. On the other hand, no BPV4 was described in the research of Gasparotto et al. [32], suggesting that this viral type was not as frequent as other types in the Amazon region. However, we were able to identify BPV4 in one flat cutaneous papilloma sample from dairy cattle, corroborating the histological diagnosis of squamous papilloma and showing the BPV4 presence at amazonian region.

In some regions, the development of neoplasms in cattle caused by these BPV types occurs, mostly, in synergism with the chronic ingestion of bracken fern (*Pteridium* spp.), due to the mutagenic substances present in the plant, as quercetin, a flavonoid that causes various DNA and chromossomal damages [2,4,8,33,34]. However, this synergism was not considered in the animals of the present study due to the need for more in-depth clinical evaluation as well as animal monitoring. Furthermore, BPV12 was isolated from a lesion classified as squamous papilloma in the study, also corroborating the literature that describes it as associated with papillomatosis in cutaneous and mucosal tissue [35,36,37,38,39]. BPV14 lesions are associated with urothelial and mesenchymal bladder tumors of cattle [33] papillomatosis and cutaneous fibropapillomatosis [2,33], and it was first described in a feline sarcoid [40]. In our study, this finding was consistent with histopathological diagnosis of squamous papilloma and fibropapilloma in a flat lesion.

In this study, the most frequent genotypes found were BPV2 (58.97%) and 13 (17.96%), which belong to the *Deltapapillomavirus* genus. Viruses of this genus are abundantly detected in cattle and, although papillomaviruses are considered strictly tissue and species-specific, *Deltapapillomaviruses* are a well-documented in cross infections into the Ruminantia clade as well as Equidae clade [2,17]. Most *Deltapapillomavirus* are etiological agents of cutaneous papillomatosis in cattle, yaks, and giraffes, as well as urinary bladder neoplasms in cattle and buffaloes, and sarcoids in equines and felines [2,5,6,33]. Additionally, this genus was already detected in the Bovidae family worldwide in a variety of biological samples such as cutaneous and fibropapillomas, blood, semen, and urine, as well as in flies [2].

Additionally, the upper GI cancer and urinary bladder tumors on these hosts are caused by chronic ingestion of bracken fern and *Deltapapillomaviruses* infection, which encodes on their genome all the oncogenes (E5, E6 and E7) [2,4]. These oncogenes are involved in viral proliferation and the host cell transformation process [4,6,17] and are considered high-risk BPVs [2]. Among the BPVs of this genus, BPV1 and 2 are the most frequently identified in the world [27,37], however, BPV1 was not detected in this study.

Most of the lesions were histologically characterized as fibropapillomas (58.33%), while squamous papillomas represented a smaller percentage (39.58%). Fibropapillomas, which consist of the proliferation of epithelial and dermal components, are lesions typically caused by BPVs of the *Deltapapillomavirus* genus [6,8,16]. In this sense, our results corroborate the literature, since the strains that were grouped in this genus (BPV2, 13, and 14) presented, for the most part, typical fibropapilloma histological characteristics (79.31%). Furthermore, it was possible to establish a predictive relationship between the occurrence of BPV2 and fibropapillomas (*p* = 0.023).

There was no significant association between the types of BPV and the occurrence of lesions in specific anatomical sites, as previously demonstrated [38]. Since the virus can be found in healthy skin and infects the basal layer after tissue microlesions, bovine cutaneous warts are likely to occur in regions of the body exposed to the causative agent of the microlesion [2,8].

The macroscopic characteristics of cutaneous papillomas are variable in cattle. The lesions may be gray to black in color, have a dry and cornified surface, appear multiple or isolated, and measure from millimeters to several centimeters. Morphologically, papillomas are described as having a cauliflower appearance, pedunculated, flat, filamentous, or rice-grain shaped [16], but there is no established relationship between macroscopy and viral type [38]. This study found that a potential trend toward association was observed between the occurrence of BPV13 and cauliflower-shaped lesions (*p* = 0.008). We described the flat lesions caused by BPV2, BPV4, BPV5, BPV12, BPV14, BPV15, and the three putative new types. Other studies [14,32] that also classified gross morphology could link papilloma lesions as pedunculated, flat, filiform, and nodular related to BPV2. Herein, it was possible to describe pedunculated, cauliflower, and flat lesions related to BPV2. Flat lesions were also linked to BPV12 in this study, while other studies classified as pedunculated [14,27,36]. However, it is difficult to indicate the relevance of the gross morphological aspect in BPV infections, and further research is needed.

Coinfections between BPVs are widely reported in cattle [12,13,29,35,37] and are associated with chronic infections, especially in cases of immunosuppression [12,41]. In this study, concomitant infection by BPVs of the *Deltapapillomavirus* genus was detected, in addition to different viral types present in the herds analyzed. BPV2, present in cases of concomitant infections observed in the study, has already been proposed as an infection facilitator by other BPVs [42]. Herein, we reported concomitant infections in animal (BPV2 and BPV13) and at the same farm (property X: BPV13 and PNT; property XIII: BPV2, 14, and PNT; property XIV: BPV2, 4, 12, and 14). Despite of the great diversity found in this study, even at a same locality and same cattle heard, the real diversity of the BPV is just an estimate. Although the FAP59 and FAP64 degenerate primer pair are able to amplify a broad range of humans and animals’ papillomavirus [3,19], there are studies demonstrating a myriad of new BPVs that go undetected in papilloma lesions [13,35]. One possible approach to detecting more than one viral strain in the same PCR-amplified sample would be to clone the amplification products, which we plan to use in future studies, along with the inclusion of internal PCR controls.

High-throughput sequencing has facilitated simultaneous identification and sequencing of multiple BPV types, revolutionizing BPV research as well as evidencing the presence of a large number of cases of co-infections [13,35]. Using this approach, 14 new viral types have recently been described and classified into the Xi, Epsilon, and *Dyoxipapillomavirus* genus, as well as an unclassified genus (BPV34) [35]. The identification of PNT types emphasizes the wide diversity of this group of viruses and the need for ongoing surveillance to identify emerging variants of potential clinical importance.

The BPV genetic material was not amplified in three samples of the present study; however, only one of these did not show characteristics of papillomatous lesions on histopathological examination. In addition to the limitations of PCR and primer specificity [2,43,44], differential diagnoses should be considered for lesions that did not present characteristics of papilloma on histology and/or for which partial L1 of BPV was not obtained. For example, orthopoxvirus infections [45], dermatophilosis, and unspecific chronic inflammation can produce crusted lesions with hyperplasia and hyperkeratosis in cattle [18].

Corroborating studies that provide information on the diversity of papillomaviruses in the Amazon region [13,14,30,31,46], this paper presents the BPVs circulating in an unstudied region, the Vale do Guaporé, Rondônia. The seven viral types and three PNT detected highlight the region’s potential for hosting unknown viral types, which reinforces the need for more studies on the clinicopathological characterization of BPV in the region. The presence of different BPV types among individual bovines and on various farms within the same geographic area prompts inquiries into viral competition, recombination, and possible interaction with other infectious agents.

The many years of animal and human PV studies and approaches, as well as the solid documentation of cross-infections between distinct clades and species and the phylogenetic analysis of the family members, have led to the idea that supports that the papillomavirus precursor was primarily skilled to infect different ecological niches on the host [45]. A thorough investigation encompassing the molecular, histopathological, and epidemiological aspects is essential to better clarify the bidirectional impact of BPV infections on bovine health and productivity. It is essential to conduct studies especially considering the zoonotic potential of a variety of papillomaviruses, since it is well known that some high-risk BPV types are able to cross-infect other species, as well as encode oncogenic properties.

Remarkable efforts have been made throughout the last decade to identify new animal papillomaviruses, especially bovine papillomavirus in Brazil, along the same lines of HPV. Molecular characterization of these viruses is important for understanding their epidemiology and pathogenicity. BPVs have frequently been reported from various parts of the world [2,17,27], and the occurrence could be influenced by geography. The presence of BPV4, BPV12, BPV14, and BPV15 in the Amazon region extends the epidemiological knowledge of the distribution of these viral types and indicates that new studies should be conducted to assess their possible association with regional environmental determinants, such as climate, genotypic diversity of host populations, and ecological interactions.

The identification of putative new viral types (PNTs) in this study adds valuable insights into BPV diversity and its potential clinical implications. Future studies employing advanced sequencing techniques, such as whole-genome sequencing, will be essential to fully characterize these PNTs. A more comprehensive genomic analysis would enable the construction of robust phylogenetic trees, refining BPV classification by improving the understanding of phylogenetic relationships among viral types. Furthermore, characterizing these PNTs could offer a broader perspective on BPV evolution, transmission dynamics, and host specificity. Such research is not only fundamental for understanding viral diversity but may also contribute to the development of improved diagnostic tools and the design of more targeted vaccines for BPV prevention in cattle populations.

The influence of environmental and genetic factors on BPV susceptibility is still not fully understood and requires further research. Enhanced surveillance and metagenomic techniques may further aid the identification of new BPV types, contributing to our understanding of viral diversity and evolution. Effective prevention strategies are needed to reduce economic losses associated with BPV in the cattle industry. Therefore, it is necessary to know the BPV viral diversity to future vaccine studies.

## 5. Conclusions

In this study, we observed a potential relationship between BPV2 infection and fibropapillomas, as well as between BPV13 and the cauliflower morphology. Furthermore, we identified the presence of Deltapapillomavirus (BPV2, 13, and 14), Xipapillomavirus (BPV4, 12, and 15), and Epsilonpapillomavirus (BPV5) circulating in the studied region. Given the potential economic and animal health implications of BPV infections, integrating molecular surveillance with histopathological and clinical assessments may provide useful insights for disease monitoring. While such an approach could contribute to broader disease prevention strategies, further studies are needed to directly evaluate preventive measures and clinical interventions.

## Figures and Tables

**Figure 1 viruses-17-00719-f001:**
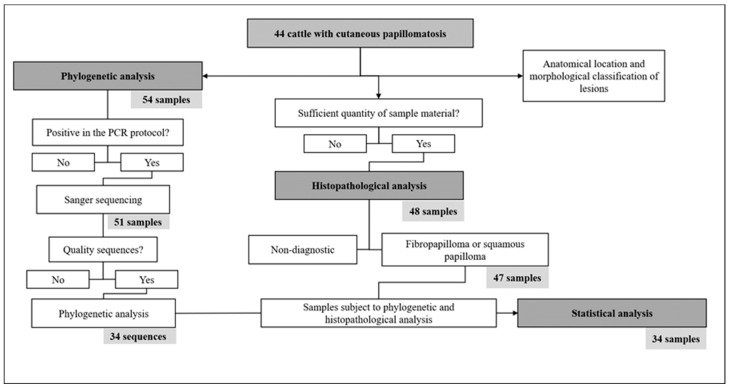
Schematic representation of the sample processing flow, including the inclusion and exclusion criteria at each analytical step.

**Figure 2 viruses-17-00719-f002:**
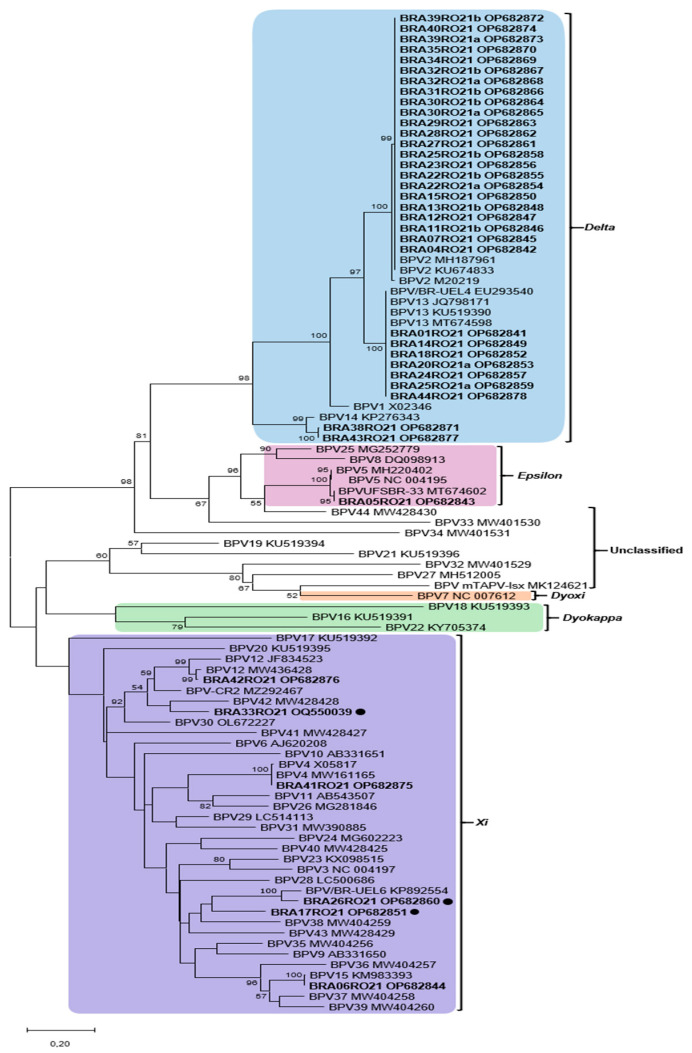
Phylogenetic tree based on the dataset of partial sequences of the BPV L1 gene. Samples from the study and the most similar sequences, as well as reference sequences from each type of BPV were included in the analysis, totalizing 95 nucleotide sequences. Evolutionary analysis was inferred using the maximum likelihood method, T92+G+I and bootstrap of 1000 replicates. The tree with the highest log likelihood (−10,731.37) is shown. The percentage at which associated taxa cluster together is shown next to the branches, bootstraps < 50% have been suppressed. The present study sequences are shown in bold; black dots indicate putative new viral types (PNT).

**Figure 3 viruses-17-00719-f003:**
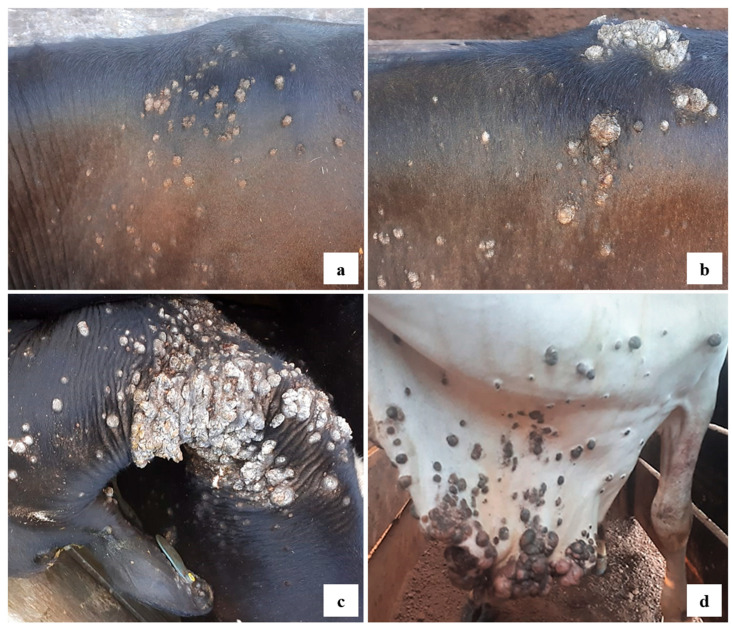
Morphology of cutaneous warts on cattle in the study. (**a**) Multiple flat papillomas on the back and thorax of animal BRA19RO21. (**b**,**c**) Exophytic masses with a cauliflower appearance on the neck and back of cattle BRA20RO21a and BRA39RO21b, respectively. (**d**) Pedunculated warts on dewlap and neck of animal BRA04RO21.

**Figure 4 viruses-17-00719-f004:**
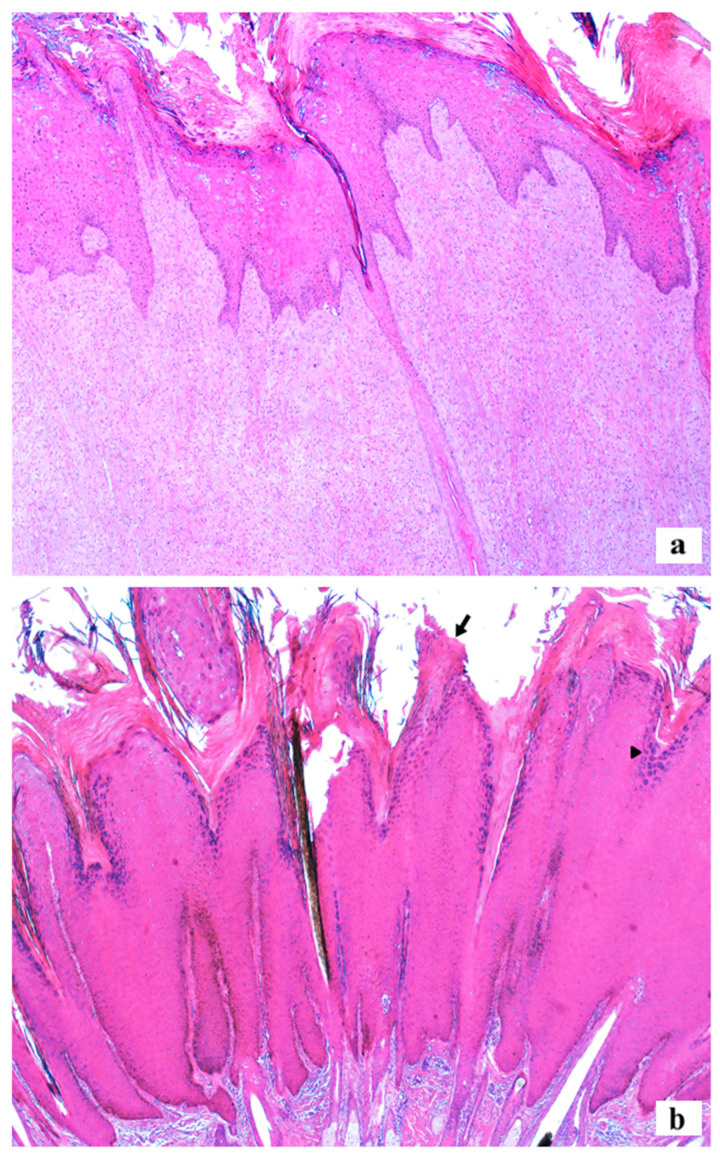
Histology of cutaneous warts from cattle sampled in the study. (**a**) Fibropapilloma: proliferation of fibroblasts in the dermis with hyperplasia and hyperkeratosis of the overlying epidermis. Haired skin, HE, 5x. (**b**) Squamous papilloma: epidermal proliferation forming exophytic projections covered by orthokeratotic hyperkeratosis (arrow). The stratum granulosum of the proliferated epidermis shows a large amount of deposition of intracytoplasmic keratohyalin granules (arrowhead). Haired skin, HE, 5x.

## Data Availability

Data are contained within the article.

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
