# Peer review of "Characterization of Papillomatous Lesions and Genetic Diversity of Bovine Papillomavirus from the Amazon Region"

_viruses, 2025, doi:10.3390/v17050719_

Round 1

Reviewer 1 Report

Comments and Suggestions for Authors

Souza et al. describe in their manuscript the collection of tissue samples from cows living in the Amazon region. 54 samples were histological classified and upon DNA extraction and papillomavirus specific PCR amplification, the partial sequence of the L1 gene was determined. The sequencing information was used to infer a phylogenetic tree. Then, they compared the newly determined sequences and the histological pictures. 

There are some points, which should be clarified prior publication:

Major concerns:

- how selective are the FAP59/64 primers? Is it possible to miss some Papillomaviruses, which are more distantly related than the detected PV. The specificity might be extrapolated by alignment of the primer sequences to the individual viral sequences or by experiments. 

- since delta BPV is the prevailing genus found in the study and these viruses are prone to infect horses as well, it would be of interest if equids are hold on farms on which delta PV were detected. 

- (line 124 to 128): There is some text ("Research manuscripts reporting large datasets that... ... prior to publication.") in the manuscript, which I think should not be there. As stated, the accession numbers of the newly determined sequences are listed in the Supplementary Table. 

- The Discussion is lengthy and has some part which do not really fit:  

(line 390): what "special environmental conditions' of Amazon region makes it an ideal place to study aspects of virological ecology"? Please explain. 

(line 399): how should the diversity of BPV in Amazon enhance diagnostic tools? 

Minor concerns:

- (Figure 1): please explain the dots and the bold font in the figure caption. Furthermore, mention and explain the scale bar. 

- (Figure 2 and 3): please indicate which samples are effectively  shown. 

- (line 45): please remove 600 Angstrom since allready mentioned before and mention the L2 protein. 

- (line 46): please change 8'000 base pairs to 8 kilo base pairs

- (line 135): please indicate the buffer of "10% buffered formaldehyde" (Tris or phosphate? molarity, pH)

- (line 350): there is a typo "knew" -> "new".

- (line 357 ff): it doesn't become clear whether a co-infection is defined as multiple virus strains detected in one animal or on one farm. Though, the detection in one animal might be excluded because of methodological limitation of sequencing from PCR amplified DNA. 

- (line 369 ff and line 417 ff): The authors mentioned high-throughput-sequencing and metagenomics as potent method to discover yet unknown (B)PV at two instances in their manuscript. However, it should not be as emphasised when they do not apply this method. 

- (line 381): differential diagnosis to PV infections is mentioned in the Discussion but no experiment is done to confirm or contradict the presence of other infections. 

Author Response

We sincerely thank Reviewer 1 for the thorough reading and for the detailed comments and suggestions. We found all your critiques, questions, and recommendations extremely helpful in improving the quality of our manuscript. Each point you raised has been carefully addressed and is explained in detail in the attached document, with the corresponding corrections made in the revised version of the manuscript.

Thank you very much for your valuable contribution to the scientific enhancement of our work.

Sincerely,
Souza et al.

Reviewer 2 Report

Comments and Suggestions for Authors

Dear authors 

I would like to thank you for the opportunity to review this manuscript, and I would also like to congratulate the authors on this study, which contributes to the understanding of BPV diversity in Brazil, particularly in the Amazon region.

Lines 124–128: This excerpt appears to have been directly copied from the journal's submission guidelines. I recommend removing it from the methodology section, as it does not provide original information related to the study. If the data have indeed been deposited, please state clearly where and provide the relevant accession numbers.

The protocol number provided (12323107.005499/2018-96) could not be located in the UFAC public research registry. Please submit the formal ethics approval document issued by the relevant ethics committee to ensure the ethical compliance of the research.

It is strongly recommended that the authors reorganize the presentation of the dataset using the 34 samples that were effectively included in the statistical analysis as the guiding reference throughout the manuscript. The current description, which references multiple sample numbers (54 collected, 44 with histology, 51 PCR-amplified, 39 sequenced, 34 analyzed), may confuse readers and obscures the study design. If the authors wish to retain these figures, a schematic presentation (such as a flowchart) is suggested to clearly outline the inclusion and exclusion criteria at each step.

In lines 131–133, the authors describe five types of lesions collected, but only three types (cauliflower, flat, and pedunculated) appear in Supplementary Table S1. Please revise this section to ensure consistency between the manuscript and the supplementary material.

I recommend to include an internal PCR control using primers for a constitutive bovine gene to ensure the integrity of theDNA. The absence of PCR amplification in some samples was attributed to technical limitations or the presence of other pathogens. However, without an internal control, DNA degradation, inefficient extraction, or PCR inhibition cannot be ruled out.

The viral genotyping was performed using FAP primers directly on PCR products. However, co-infections with multiple BPV types are common in bovine papillomatosis lesions, and direct sequencing may produce chromatograms with overlapping peaks, hindering sequence readability. Have the authors considered the potential for co-infections in these samples? How might these co-infections affect the observed morphological and histopathological features of the lesions? For samples where FAP sequencing was unsuccessful, cloning the PCR products might have enabled recovery of different viral types from the same lesion, increasing sequencing success and allowing a more comprehensive assessment of viral diversity. I recommend that this limitation be discussed, and, if feasible, cloning should be considered in future analyses—or even in this study.

The analysis involving 34 samples revealed associations between BPV2 and fibropapillomas (p = 0.023), and between BPV13 and cauliflower-like lesions (p = 0.008). While the statistical methods are appropriate, the small sample size and especially the low frequency of certain viral types limit the robustness of these conclusions. Furthermore, no correction for multiple comparisons appears to have been applied, increasing the risk of type I errors (false positives). I recommend that these findings be described as exploratory or hypothesis-generating, and that the manuscript language be adjusted. For example, instead of stating “an association was found,” consider using “a potential trend toward association was observed” .

The discussion includes an extended section on BPV4, despite it being detected only once. It would be advisable to present this information more concisely and maintain focus on the broader trends, particularly the relative frequency and diversity of viral types identified in the Amazon region.

It would be valuable for the authors to discuss the importance of the PNTs identified in this study and their potential implications for future BPV classification. This could further emphasize the significance of the findings and their contribution to viral diversity research.

In the conclusion section, I suggest that the authors make some adjustments. These findings are better framed as exploratory observations, rather than definitive conclusions. The recommendation to integrate molecular, histopathological, and clinical assessments for disease monitoring and prevention is reasonable. But this study did not directly evaluate preventive strategies or clinical interventions, so this part of the conclusion would benefit from a more cautious and contextualized phrasing.

Author Response

Dear Reviewer,

We sincerely thank you for your thorough review and constructive feedback. Your comments, suggestions, and criticisms were extremely helpful and contributed significantly to the improvement of our manuscript.

Please note that each of your points has been addressed in detail in the attached response document, where we explain the changes made and provide the necessary clarifications.

Thank you once again for your valuable contribution to this study.

Kind regards,

Souza et al

Round 2

Reviewer 2 Report

Comments and Suggestions for Authors

All recommendations regarding the manuscript were previously submitted, and the authors have satisfactorily revised the manuscript.